# Human papillomavirus genomes associate with active host chromatin during persistent viral infection

Alix Warburton[1], Tovah E. Markowitz[2], JJ L. Miranda[3], Kinjal Majumder[4,5], Alison A. McBride[1]*

1 Laboratory of Viral Diseases, National Institute of Allergy and Infectious Diseases, National Institutes of Health, Bethesda, Maryland, United States of America, 2 Integrated Data Sciences Section, Research Technologies Branch, National Institute of Allergy and Infectious Diseases, National Institutes of Health, Bethesda, Maryland, United States of America, 3 Department of Biology, Barnard College, Columbia University, New York, New York, United States of America, 4 Institute for Molecular Virology, University of Wisconsin-Madison, Madison, Wisconsin, United States of America, 5 McArdle Laboratory for Cancer Research, University of Wisconsin School of Medicine and Public Health, Madison, Wisconsin, United States of America

* amcbride@nih.gov

## Abstract

Human papillomaviruses (HPVs) maintain their genomes as minichromosomes in the nuclei of infected keratinocytes. This study investigates the association of HPV31 genomes with host chromatin using both HiC and 4C-seq chromosome conformation capture techniques. We show that HPV31 genomes preferentially associate with transcriptionally active A compartments of host chromatin, regions of open chromatin defined by ATAC-seq, and super-enhancers defined by Brd4 and H3K27ac ChIP-seq. The viral genome association sites were also highly correlated with genomic loci previously identified as common HPV integration sites in cervical cancers. Recent studies have shown that transcriptionally active sites are prone to dsDNA breaks, and we find a strong correlation among dsBREAK datasets with transcriptionally active and open regions of host chromatin and the HPV31 genome association sites defined in our study. These findings suggest that HPV genomes associate with cellular transcriptional epicenters to maintain active viral gene expression during persistent infection, but also indicate that the susceptibility of these regions to dsDNA breaks could explain their propensity for viral DNA integration in HPV-associated cancers.

## Author summary

HPV minichromosomes can sustain long-term, persistent infections within the basal keratinocytes of stratified epithelia. Notably, persistent infection with on-cogenic HPV types accounts for ~5% of human cancers. To better understand the mechanism of viral persistence, we mapped global HPV31-host chromatin

**Data availability statement:** The data that support the findings of this study are publicly available from: GEO accession series GSE294036 , GSE294064 and GSE294037. All code used for data analysis can be found at https://github.com/OpenOmics/Markowitz_HPV_CCC_analysis.

**Funding:** This research was supported by the Division of Intramural Research at the National Institute of Allergy and Infectious Diseases, part of the Intramural Research Program of the National Institutes of Health (NIH), NIH grant number ZIA AI001223 and ZIA AI000713 to AAM. The contributions of the NIH authors are considered Works of the United States Government. The findings and conclusions presented in this paper are those of the authors and do not necessarily reflect the views of the NIH or the U.S. Department of Health and Human Services. https://www.niaid.nih.gov/about/dir.

**Competing interests:** The authors have declared that no competing interests exist.

interactions in cervical keratinocytes. We show that HPV31 minichromosomes preferentially associate with host euchromatin and sites that frequently contain integrated HPV DNA in cervical cancers. We further demonstrate a strong correlation between cell-type specific double-stranded DNA breaks with transcriptionally active and accessible regions of host chromatin and the HPV31 genome association sites delineated in our study. These findings suggest that HPV genomes target cellular transcriptional hubs to maintain viral gene expression during persistent infection. The susceptibility of these same regions to DNA breaks may also explain why viral DNA frequently integrates into these sites in HPV-associated cancers.

## Introduction

Human papillomaviruses (HPVs) replicate in stratified epithelia and their genomes are maintained as low-copy number minichromosomes within the nuclei of the dividing basal keratinocytes. Papillomavirus genomes associate with host mitotic chromosomes to ensure partitioning to daughter cells during maintenance replication [1], and the E2 proteins from several papillomaviruses have been shown to tether viral genomes to mitotic chromatin [2,3]. Several cellular factors have been implicated in this tethering mechanism including Brd4, which colocalizes with many papillomavirus E2 proteins on mitotic chromosomes [4–6], and binds to sites of euchromatin within the host genome [7–9]. However, the exact mechanism of HPV genome partitioning, and persistence is not completely understood.

Persistent infection with oncogenic HPVs can result in accidental integration of the viral genome into host chromatin, and this is a key event in many HPV-associated cancers [10]. HPV DNA is found integrated in all human chromosomes, but often occurs within common fragile sites (difficult to replicate regions of the genome) [11–13], and/or transcriptionally active regions [14–16]. We have shown previously that sites of recurrent HPV DNA integration in different cervical tumors (integration hotspots) are enriched in large clusters of enhancer elements termed super-enhancers, some of which function as regulatory hubs for cell-identity genes [17]. These transcriptionally active regions are attractive tethering sites for HPV genomes as they could ensure active viral transcription to support viral genome persistence.

Here, we use chromosome conformation capture techniques to map the association of extrachromosomal HPV31 genomes with host chromatin in cervical keratinocytes. We identified regions of HPV31 genome association with host chromatin using both HiC and 4C-seq technologies and characterized the genetic and epigenetic features of these sites. Regions of HPV31 genome association were overrepresented in transcriptionally active A compartments, enriched at super-enhancers, and at sites previously shown to be HPV integration hotspots [17]. We propose that HPV genomes associate with active chromatin within host nuclei to ensure an active infection during viral persistence.

## Results

### Mapping HPV31-host *trans*-interactions in cervical keratinocytes by HiC

We previously performed HiC experiments in two cervical keratinocyte cell lines that contain extrachromosomal HPV16 (W12-20863) or HPV31 (CIN612-9E) genomes as a comparison with Burkitt lymphoma cell lines containing extrachromosomal Epstein-Barr virus (EBV) genomes [18]. EBV genomes were found to preferentially associate with gene-poor chromosomes during latency, and gene-rich euchromatin during lytic reactivation. In contrast, HPV genomes did not show a chromosomal preference based on gene-density in cell lines that contained extrachromosomal HPV16 or HPV31 genomes [18]. To investigate the regions of association between HPV and host chromatin in more detail, we sequenced a subset of the HiC dataset from CIN612-9E (9E) cells [18], to a sequencing depth of more than three billion 150 base paired-end reads per sample. A schematic representation of the HiC technique is illustrated in Fig 1A. HiC contact maps were generated for 1 Mb bins using HiC-Pro [19], and significant *trans*-interactions identified using the Multi-function Hi-C data analysis tool (MHiC) [20]. This identified a total of 17,456 and 236,546 interactions for Replicates 1 and 2, respectively, and these represent a combination of host-host, viral-viral, and viral-host contacts. As commonly reported [21], we found strong interchromosomal interactions between the small, gene-rich human chromosomes 16, 17, 19 and 22 in both HiC datasets, and enrichment of HPV genomes with chromosomes 1, 2 and 19 as observed in our previous study [18], Fig 1B-D.

Next, viral-host *trans*-interactions were identified as significant 1 Mb bins with a q-value of <0.1. This resulted in a total of 92 bins containing HPV31 association domains within the human genome across the two replicates (S1 Table). Significant viral-host *trans*-interactions from the two HiC datasets were combined and overlapping regions consolidated. This resulted in a total of 87 bins containing HPV31 association sites; these were used for downstream analyses and are listed in S1 Table. The location of these regions within the host genome is illustrated in Fig 1D and shows at least one viral association domain on chromosomes 1–3, 6–8, 10–12, 16–17 and 19, with eight bins common (directly overlapping or abutted) to both HiC samples (Fig 1D and S1 Table).

A higher number of total and *trans*-interactions were mapped in HiC Replicate 2, compared to Replicate 1. It has been shown that chromosome topology is very dynamic and can change considerably during the cell cycle [22,23] and even after continued passage of cells [24]. In our case, care was taken to use similar early cell passes for each replicate, and to harvest cells at similar confluency (which should affect cell cycle distribution). However, the difference in the number of interactions could have been influenced by the fixation conditions used; Replicate 1 and 2 cells were crosslinked *in situ* or in suspension, respectively.

### Analysis of host attachment sites in the HPV31 genome

In addition to identifying regions of host chromatin associated with viral minichromosomes, HiC can map the precise region of the viral genome that mediates this association. Our HiC procedure used MboI to digest viral and host chromatin, and the distribution of these sites across the HPV31 genome is shown in S1 Fig. Thousands of viral-host contacts were detected at most sites across the genome, with the highest number located in the early region. However, due to the limited size of the HPV genome, we conclude that it is challenging to accurately map viral-host chromatin interactions using HiC.

### Mapping HPV31 viral-host interactions by 4C-seq in 9E cells

To detect higher resolution viral-host *trans*-interactions in the host genome, and to compare with viral genome association domains identified by HiC, we performed 4C-seq in CIN612 cells containing extrachromosomal (9E cells) or integrated (6E cells) HPV31 DNA using the viral genome as bait. The 4C sequencing process involves crosslinking cells, followed by restriction digestion, and proximity ligation of neighboring chromatin fragments. The viral-host fusion fragments were

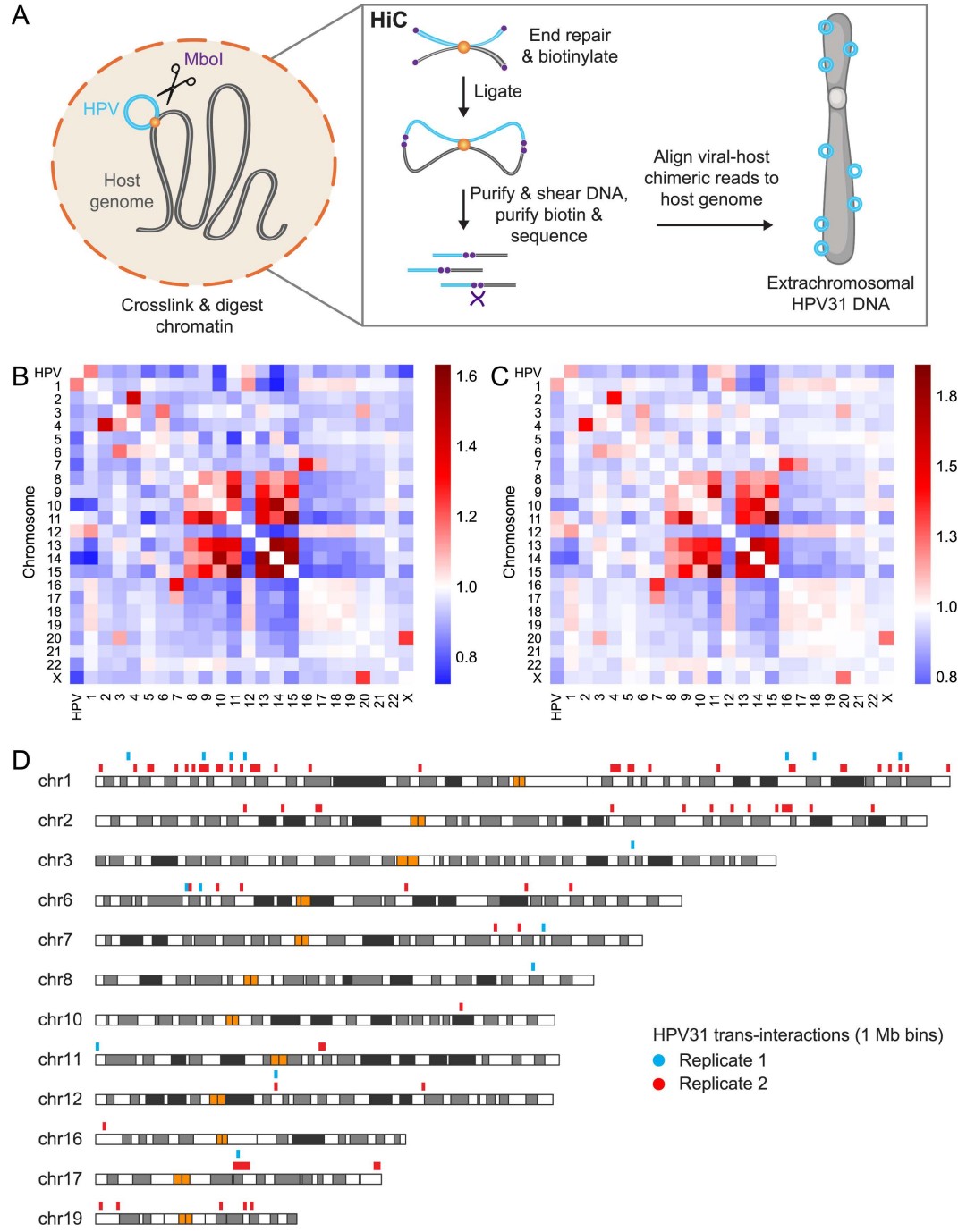

**Fig 1. Mapping HPV31-host chromatin interactions by HiC.** A, Schematic representation of the *in situ* HiC methodology. B-C, Heatmaps showing interchromosomal contacts between the HPV31 and human genomes for the Replicate 1 (B) and Replicate 2 (C) 9E HiC datasets. Scale bars represent enrichment (red) or depletion (blue) of observed/expected counts. D, Alignment of HPV31 genome association domains (1 Mb bins, S1 Table) mapped by HiC in 9E cells with host chromosomes. Blue (Replicate 1) and red bars (Replicate 2) represent viral-host *trans*-interactions identified from two biological replicates, and orange bars represent centromeres. Parts of the figure were created in BioRender, https://BioRender.com/4bm1tx9.

amplified by inverse PCR for unbiased detection of all chromosome interactions with a particular genomic region of interest (viewpoint), in our case the HPV31 genome. A schematic representation of the 4C method is illustrated in Fig 2A. We used EcoRI as the primary restriction enzyme due to the limited viewpoint selection in the small HPV31 genome and the success of enzymes with 6 bp recognition sites (including EcoRI) in previous studies of viral-host interactions with other small DNA viruses [25–28]. EcoRI cuts the HPV31 genome at nucleotide positions 3,362 (E2 and E4 ORFs) and 6,128 (L1 ORF). DpnII was used as the secondary restriction enzyme, which cuts at the same nine restriction sites within the HPV31 genome as MboI used for the HiC experiments (Fig 2B). Inverse PCR primers (S2 Table) were designed at the boundaries of each restriction fragment (viewpoint) generated by neighboring EcoRI and DpnII sites. Only viewpoints that

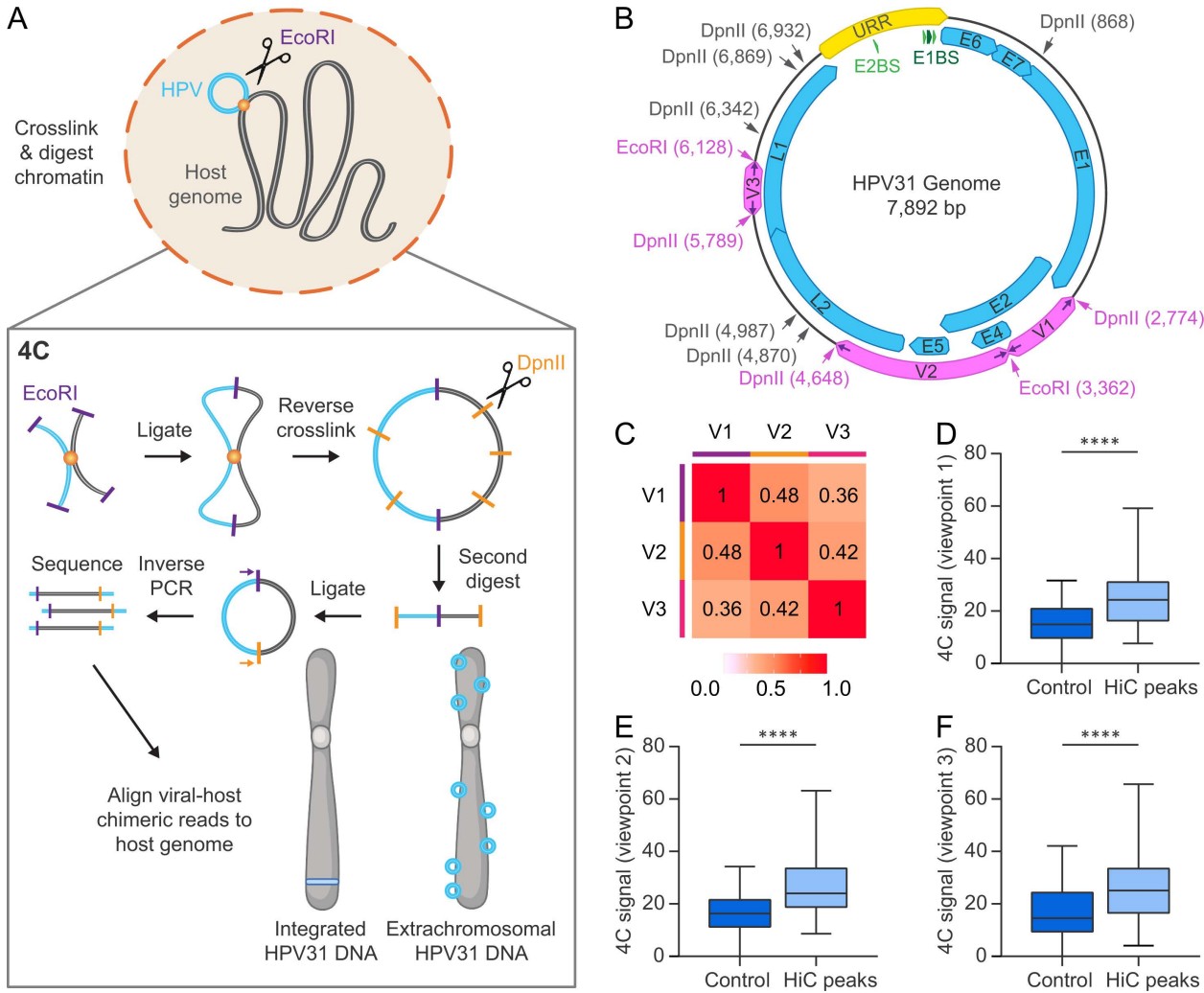

**Fig 2. Analysis of HPV31-host genome association sites by 4C-seq.** A, Schematic representation of the 4C-seq methodology. B, HPV31 genome showing restriction sites used for 4C (EcoRI, primary restriction enzyme; DpnII, secondary restriction enzyme). MboI, which shares the same recognition sequence as DpnII, was used for HiC. Inverse PCR primers (indicated by purple arrows) targeting three viewpoints (V1-V3) within the HPV31 genome were used for 4C analysis; the corresponding primary and secondary restriction sites targeted by the 4C primers are indicated by pink text. C, Heat-map showing Pearson correlation coefficients of 4C signals (average normalized reads across all four 9E samples) for 50 kb windows across the three viewpoints. D-F, Boxplots showing 4C signal intensities (average normalized reads across all four 9E samples for 50 kb windows) at HiC peaks relative to size-matched random control regions for viewpoints 1 (D), 2 (E) and 3 (F) on the HPV31 genome. Wilcoxon rank-sum test was used to determine statistical significance (****, p < 0.0001). Parts of the figure were created in BioRender, https://BioRender.com/4bm1tx9.

exceeded 300 bp were selected for 4C analysis, based on a minimum viewpoint size to ensure efficient recircularization during the second ligation step [29]. This resulted in a total of three HPV31 viewpoints that encompassed the E2 and E4 genes (viewpoint 1), E2, E4, E5 and L2 genes (viewpoint 2), and L1 gene (viewpoint 3), Fig 2B. For each of the seven samples listed in S3 Table, PCR products generated from the three viewpoints were sequenced on the Illumina NovaSeq 6000 platform. The number of reads per sample are detailed in S3 Table.

Viral-host genome contact profiles were mapped for the three viewpoint reading primers using the pipe4C pipeline [30]. Pairwise interaction frequencies of ligation events measured by 4C-seq are semi-quantitative and subject to experimental biases. To mitigate these biases, and reduce individual fragment noise, normalized contact frequencies were averaged across 50 kb genomic windows (based on an approximate number of 12 EcoRI restriction sites per window). To assess whether the three HPV31 viewpoints identified similar or distinct regions of host chromatin, we calculated the average normalized counts (50 kb windows) for the four 9E replicates per viewpoint and compared them using the Pearson correlation coefficient. This showed a moderate degree of overlap among 4C signals associated with the three viewpoints (Fig 2C). The normalized read coverage (4C signal) of each viewpoint dataset was also compared to the HPV31 *trans*-interaction domains identified by HiC. This showed that the 4C signal for all three viral viewpoints was higher at HiC peaks compared to random genomic control regions, Fig 2D-F. These data suggest that the association of the viral genome with host chromatin is not limited to a specific region of the viral genome, as we concluded above for HiC, but again the small size of the viral genome and resolution of this technique makes it difficult to determine the precise location of attachment with host chromatin in the viral genome.

## Comparison of HPV31 viral-host interactions in cells with integrated or extrachromosomal viral genomes

Integrated HPV genomes can make short- and long-range *cis*-interactions with host chromatin [31]. Therefore, both as a positive control for the 4C assay, and to monitor the level of background reads across the host genome, we performed three biological replicates of 4C-seq in the 6E clone of CIN612 cells, which contains approximately two to five copies of integrated HPV31 DNA [32] on chromosome 4. Viral-host chimeric reads were identified using the pipe4C pipeline [30]. However, viewpoint 1 and 2 primers did not give the expected reads because integration of the HPV31 genome in 6E cells had disrupted the targeted region. This was confirmed by PCR analysis, which identified large deletions within the viral E1, E2, L2 and L1 genes (S2 Fig).

Nevertheless, as predicted, alignment of viewpoint 3 sequencing reads showed strong *cis*-interactions on chromosome 4 at the site of viral DNA integration (Fig 3). However, low levels of *trans*-interactions between the integrated HPV31 genome and host chromosomes were also detected across all chromosomes in 6E cells; we assume that some of these represent background reads inherent to the 4C procedure but could also sometimes represent bone fide interchromosomal interactions. These putative background regions were identified for all chromosomes (except chromosome 4) using the method outlined below for 9E cells, and these regions are listed in S4 Table. However, this background subtraction strategy could only be used for viewpoint 3 reads and so the downstream correlation analyses shown below focused only on the viewpoint 3 dataset.

## HPV31 Viral-host *trans*-interactions mapped by HiC and 4C-seq are highly correlated

Next, significant 4C peaks of interaction were defined in 9E cells based on a method previously used to map EBV attachment sites within the host genome [33]. Sequencing data includes random noise, which can typically be modeled by a Poisson distribution. The peak calling method described by Kim *et al.* uses this approach to identify regions with significantly higher read counts than expected by chance, suggesting biological signals. We therefore applied a Poisson distribution model across all 6E and 9E 4C viewpoint 3 datasets to identify background peaks in the 6E control samples. Significant viral-host *trans*-interactions were then identified using methods previously described [33]. Briefly, reads generated from the viewpoint 3 reading primer were aligned to the hg38 reference genome and

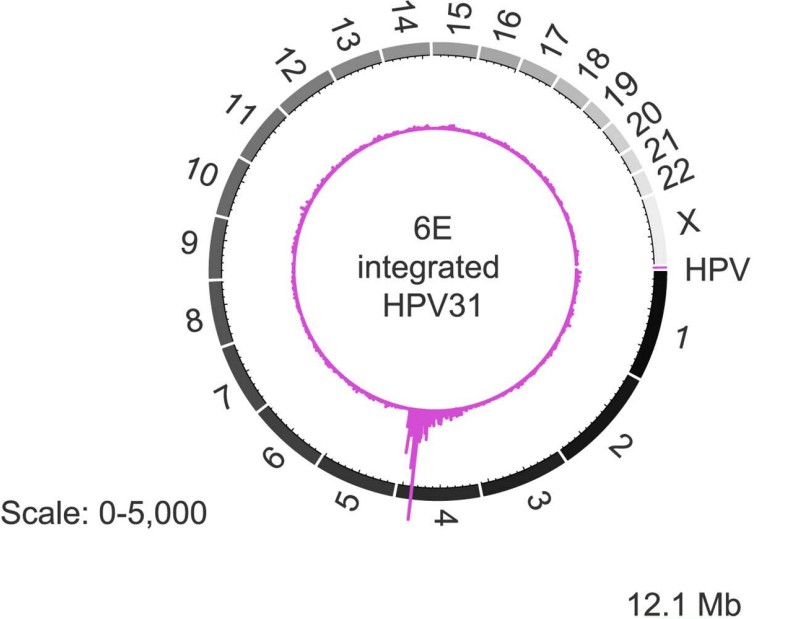

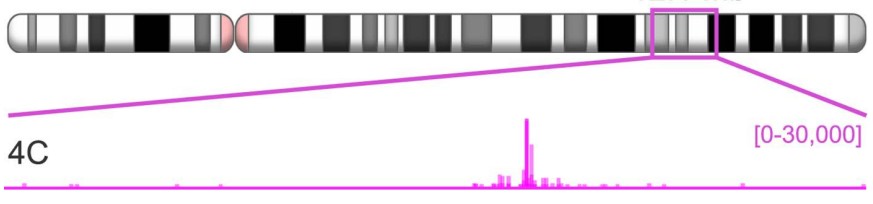

**Fig 3. 4C-seq analysis in 6E cells containing integrated HPV31 DNA.** CIN612-6E cervical keratinocytes that contain integrated HPV31 DNA on chromosome 4 were used as a positive control for the 4C-seq assay. Alignment of 6E 4C sequencing reads associated with viewpoint 3 (purple peaks) to host chromosomes (*top*) and chromosome 4 (*bottom*). Prominent 4C peaks mark the HPV31 integration site. Numbers in parentheses indicate the range of average normalized 4C reads for 50 Kb windows. No 4C output files were generated for viewpoint 1 and 2 in 6E cells due to deletions within the integrated HPV31 genome at regions targeted by the associated inverse PCR primers (S2 Fig).

assigned into non-overlapping 50 kb windows. P-values were assigned using a Poisson distribution model and significant regions defined as windows with a p-value < 10e-5. Neighboring windows were then merged into a single bin based on a maximum gap size of 50 kb and a minimum length of 100 kb. As described above, only viewpoint 3 samples were used for subsequent analyses, as we could assess 4C background regions for this viewpoint from 6E control cells. Between 1,564 and 2,005 significant regions were identified for each of the four 9E samples for viewpoint 3 and are listed in S5 Table.

We further defined consensus peaks for viewpoint 3 as significant *trans*-interactions found in three out of four 4C datasets, which resulted in 371 HPV31 "consensus" association domains (S6 Table). The genomic location of these 4C consensus peaks is shown in Fig 4A. Next, we intersected HiC peaks and 4C consensus peaks and found 19 regions in common, with 28 4C peaks contributing to these overlaps (multiple 4C peaks sometimes aligned with a single HiC peak), Fig 4B. Notably, the MYC, E2F2 and CASP8 genes, which are important regulators of the cell cycle and apoptosis and have previously been implicated in HPV-related processes or cancer [34–36], were identified as common *trans*-interaction sites in both the HiC and 4C datasets (Fig 4C). Permutation testing showed that *trans*-interactions identified from the two methods were significantly correlated (p-value, 1.0E-4). We conclude that extrachromosomal HPV genomes are enriched at specific genomic locations.

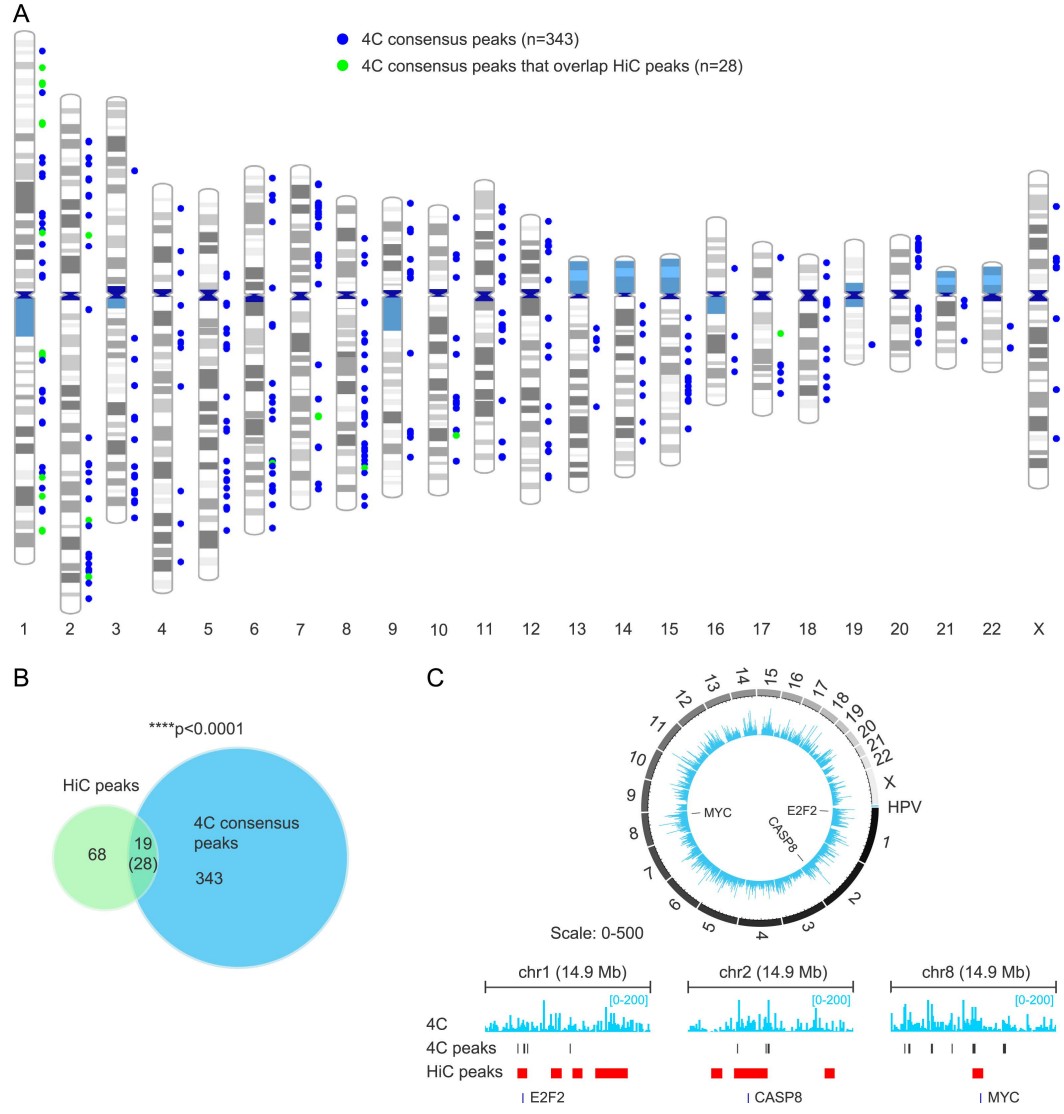

**Fig 4. High correlation of HPV31 *trans*-interactions mapped in 9E cervical keratinocytes by HiC and 4C-seq.** A, Alignment of 4C consensus peaks (viewpoint 3) to host chromosomes. Colored circles represent different viral-host *trans*-interactions that overlap (green) or do not overlap (blue) HiC peaks. B, Venn diagram showing the overlap of viral-host *trans*-interactions identified by HiC (n = 87 peaks) and 4C-seq (n = 371 consensus peaks). The number in parentheses (28) indicates 4C peaks that overlap HiC peaks; in some instances, multiple 4C peaks overlapped a single HiC peak giving rise to the lower number (19). Permutation testing showed significant overlap between HiC and 4C peaks (p < 0.0001). C, Alignment of 4C sequencing reads from 9E cells associated with viewpoint 3 (light blue peaks) to host chromosomes (*top*) and chromosomes 1 (chr1:19,916,514-34,853,899), 2 (chr2:194,879,295-209,756,896) and 8 (chr8:119,948,588-134,825,931) at the E2F2, CASP8 and MYC gene loci, respectively (*bottom*). HiC peaks and 4C consensus peaks are denoted by red and dark grey bars, respectively. Numbers in parentheses indicate the range of average normalized 4C reads for 50 Kb windows.

## Extrachromosomal HPV31 genomes preferentially associate with active A compartments of host chromatin

In eukaryotic nuclei, chromosomal regions are functionally and spatially organized into two mutually exclusive compartments; active (A) euchromatin and repressive (B) heterochromatin. To determine whether HPV31 genome attachment sites are enriched in either compartment, we defined A and B compartments in our two HiC 9E cell datasets using HiTC

software [37]. In this analysis, 9E HiC contact matrices were binned into 0.5 Mb windows and assigned to A or B compartments based on principal eigenvector analysis. Specifically, positive eigenvalues were associated with the A compartment due to a higher gene density in those regions. A total of 4,968 bins were defined of which 2,144 (43.2%) were designated A compartments in both HiC datasets (AA), 2,564 (51.6%) were designated as B compartments (BB), and 260 (5.2%) were called A in one dataset and B in the other (AB), Fig 5A and S7 Table. The proportion of the 9E genome that was classified into the A compartment (~46%) was similar to that reported for HeLa (57%; Jaccard score 0.55) and Normal Human Epidermal Keratinocytes (NHEK) cells (59.2%; Jaccard score 0.58) [38].

HPV31 *trans*-interactions identified by HiC and 4C were intersected with the A/B compartments defined from the two HiC datasets, Fig 5B-D. For the HiC dataset, 79.1% *trans*-interactions were associated with active A compartments defined in both datasets (AA), 17.4% associated with repressive B compartments (BB), and 3.5% spanned both A and B compartments (both), (Fig 5B). Consistent with the HiC dataset, 85.7% HPV31 *trans*-interactions identified from 4C-seq localized to A compartments (AA), 10.2% with B compartments (BB), 3.8% with A compartments defined in only one HiC dataset (AB), and 0.3% spanned the transition between A and B compartments (both), Fig 5C. Permutation testing

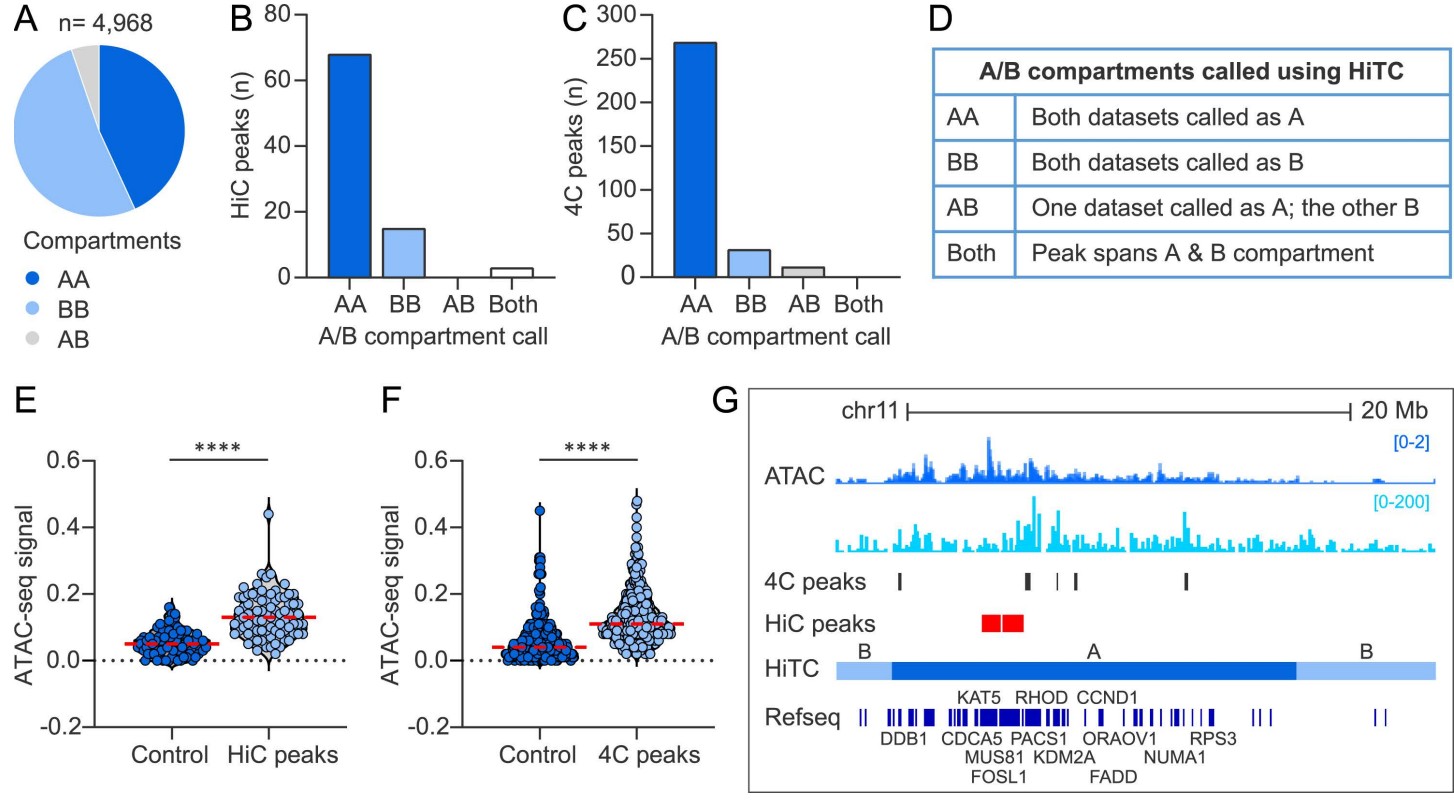

**Fig 5. Extrachromosomal HPV31 genomes preferentially associate with active regions of host chromatin.** A/B compartments were defined using the two HiC contact matrices from 9E cells and correlated with viral-host *trans*-interactions mapped by HiC and 4C-seq. A, Pie chart showing the distribution of active (A) and repressive (B) chromatin compartment calls, grouped as indicated in the table in panel D. B-C, Bar charts showing the association of HiC peaks (B) and 4C consensus peaks (C) with A/B compartments grouped as indicated in the table in panel D. E-F, Violin plots showing ATAC-seq signal intensities (average reads per genomic content) profiled in growing 9E cells at HiC peaks (E) and 4C consensus peaks (F) relative to size-matched random control regions. Red dotted line indicates mean intensity of 4C signal. Wilcoxon rank-sum test was used to determine statistical significance (****, p < 0.0001). G, Alignment of viewpoint 3 4C (light blue peaks) and ATAC-seq (dark blue peaks) datasets to chromosome 11 (chr11:58,924,490-85,306,749; hg19). HiC peaks, 4C consensus peaks and genes highly expressed in cervical keratinocytes [39] are denoted by red, dark grey and dark blue bars, respectively. Genes previously noted to be associated with HPV infection and/or cancer are indicated. Active A and repressive B chromatin compartments profiled in the 9E HiC datasets using HiTC are indicated.

showed that HPV31 *trans*-interactions were significantly enriched at active A compartments for both the HiC (p-value, 1.0E-4) and 4C (p-value, 1.0E-4) datasets. These findings support that HPV31 minichromosomes target transcriptionally active regions of host chromatin during persistent viral genome replication.

## Extrachromosomal HPV31 genomes preferentially associate with open accessible chromatin

To gain further insight into the association of HPV31 genomes with chromatin, we examined the correlation between HiC peaks and 4C consensus peaks with chromatin accessibility regions mapped in 9E cells by ATAC-seq. We found that HPV31 genome association domains (listed in S1 and S6 Tables for the HiC and 4C datasets, respectively) exhibited greater chromatin accessibility than random control regions, Fig 5E-F. A comparison of HPV31 *trans*-interactions, ATAC-seq signals, and A and B compartments are shown in Fig 5G for chromosome 11 (chr11:58,924,490–85,306,749; hg19) in 9E cells. Notably, this genomic region encodes several genes that are highly expressed in cervical keratinocytes [39]. These genes are involved in the regulation of cell cycle (CCND1, CDCA5, FOSL1, KDM2A, NUMA1, RHOD), apoptosis (FADD, ORAOV1), or the DNA damage response (DDB1, KAT/TIP60, MUS81, PACS1, RPS3), and several have been implicated in the HPV lifecycle and/or HPV-associated cancers [40–47]. Collectively, these data support the hypothesis that extrachromosomal HPV31 genomes preferentially associate with host euchromatin, including gene loci that encode proteins involved in HPV infection and cancer.

## HPV31 minichromosomes associate with regions containing super-enhancers

We have previously shown that regions of recurrent HPV DNA integration which occur in many cervical cancers (integration hotspots) are enriched in super-enhancers [17] and proposed that these super-enhancers are attractive nuclear locations for HPV minichromosomes. Super-enhancers regulate genes important for cell identity and function and, as such, are highly tissue- specific [48]. We have previously mapped the super-enhancer markers Brd4 and H3K27ac in W12 HPV16-positive cervical keratinocytes by ChIP-seq and shown that >80% of enriched peaks overlapped with ENCODE defined enhancers in NHEK cells [17], indicating a high level of tissue-specificity and conservation of these regulatory elements amongst keratinocytes. To assess the association of HPV31 *trans*-interaction sites with super-enhancers, we aligned HiC peaks (1 Mb bins) with the previously mapped W12 super-enhancers (S8 Table) [17]; this showed good correlation (p-value, 1.0E-4) between the two datasets (Fig 6A-B). We also correlated the 4C viewpoint 3 and cellular super-enhancer datasets and showed that 4C signal intensities (average normalized reads for all four 9E samples across 50 kb windows) were significantly higher at these elements relative to size-matched random control regions (p-value, 2.2E-16), Fig 6C. Furthermore, bootstrap analysis showed 4C consensus peaks were significantly associated with the super-enhancers active in cervical keratinocytes (p-value, 1.0E-4), Fig 6D. We conclude that HPV31-host genome association domains are enriched at transcriptionally active regions of host chromatin enriched in super-enhancers.

To identify biological processes associated with the HPV31 genome association domains, GREAT (Genomic Regions Enrichment of Annotations Tool) analysis was performed using the 4C consensus peaks. GREAT uses a binomial test to identify enriched gene ontology terms associated with regions of interest [49]. Gene ontology analysis of our 4C dataset identified epidermis and epithelium development; epidermal, epithelial and keratinocyte cell differentiation and epithelial cell proliferation to be significantly enriched biological processes associated with regions of HPV31-host genome attachment (Fig 6E and S9 Table). This confirms that the HPV31 genomes preferentially associate with transcriptionally active regions and their associated super-enhancers, and particularly those that determine keratinocyte cell identity.

## Extrachromosomal HPV31 association sites are enriched at HPV integration hotspots

We have previously shown that sites of recurrent HPV integration in cervical tumors are also enriched in keratinocyte super-enhancers [17]. To directly examine recurrent HPV integration sites with the extrachromosomal HPV31 association regions, the HiC (S1 Table) and 4C-seq (S6 Table) datasets were examined for correlation with our previously defined

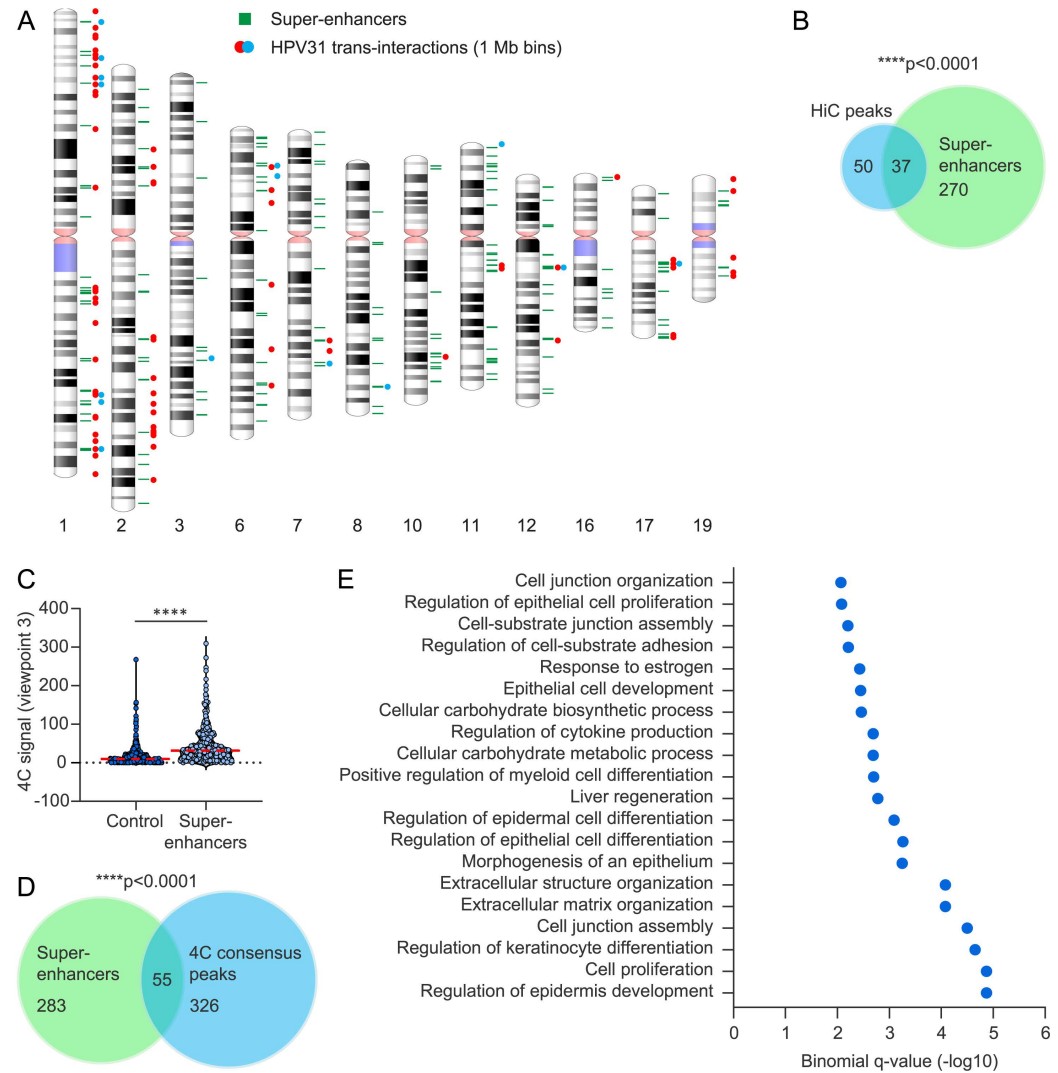

**Fig 6. HPV31-host interaction sites are enriched at cellular identify gene loci.** A, Alignment of HPV31 genome association domains (1 Mb bins) mapped by HiC in 9E cells with cellular super-enhancers. Red and blue circles represent HPV31 *trans*-interactions from two biological replicates; two circles represent regions common across the two datasets. Green bars represent super-enhancers previously mapped in HPV16-positive cervical keratinocytes (17). B, Venn diagram showing the regions of overlap between HiC peaks (n=87) and super-enhancers (n=338). C, Violin plot showing 4C signal intensities (average normalized reads across all four 9E samples for 50 kb windows) at super-enhancers relative to size-matched random control regions. Red dotted line indicates mean intensity of 4C signal. Wilcoxon rank-sum test was used to determine statistical significance (***, p<0.0001). D, Venn diagram showing the regions of overlap between 4C consensus peaks (n=371) and super-enhancers. Permutation testing showed significant overlap between HiC peaks (B) and 4C consensus peaks (D) with super-enhancers (p<0.0001). E, Enriched biological processes associated with HPV31-host genome interaction domains identified by GREAT (Genomic Regions Enrichment of Annotations Tool) analysis. Shown are the top 20 binomial enriched terms associated with 4C consensus peaks that have an adjusted p-value (false discovery rate q-value) of <0.05 and at least two-fold observed over expected enrichment (S9 Table).

sites of recurrent HPV integration (S10 Table) [17]. We found that 31/87 (35.6%) HiC peaks and 65/371 (17.5%) 4C consensus peaks overlapped and showed significant enrichment at these integration hotspots (p-value, 1.0E-4 and 7.0E-4, respectively), Fig 7A. This represents 24/36 (66.7%) integration hotspots (Fig 7B and S10 Table). Consistent with extra-chromosomal HPV association regions, HPV DNA integration breakpoints occurred more frequently in active chromatin compartments defined by HiC in 9E cells (AA, 66.2%; BB, 31.9%; AB, 1.9%), Fig 7C, and permutation testing showed

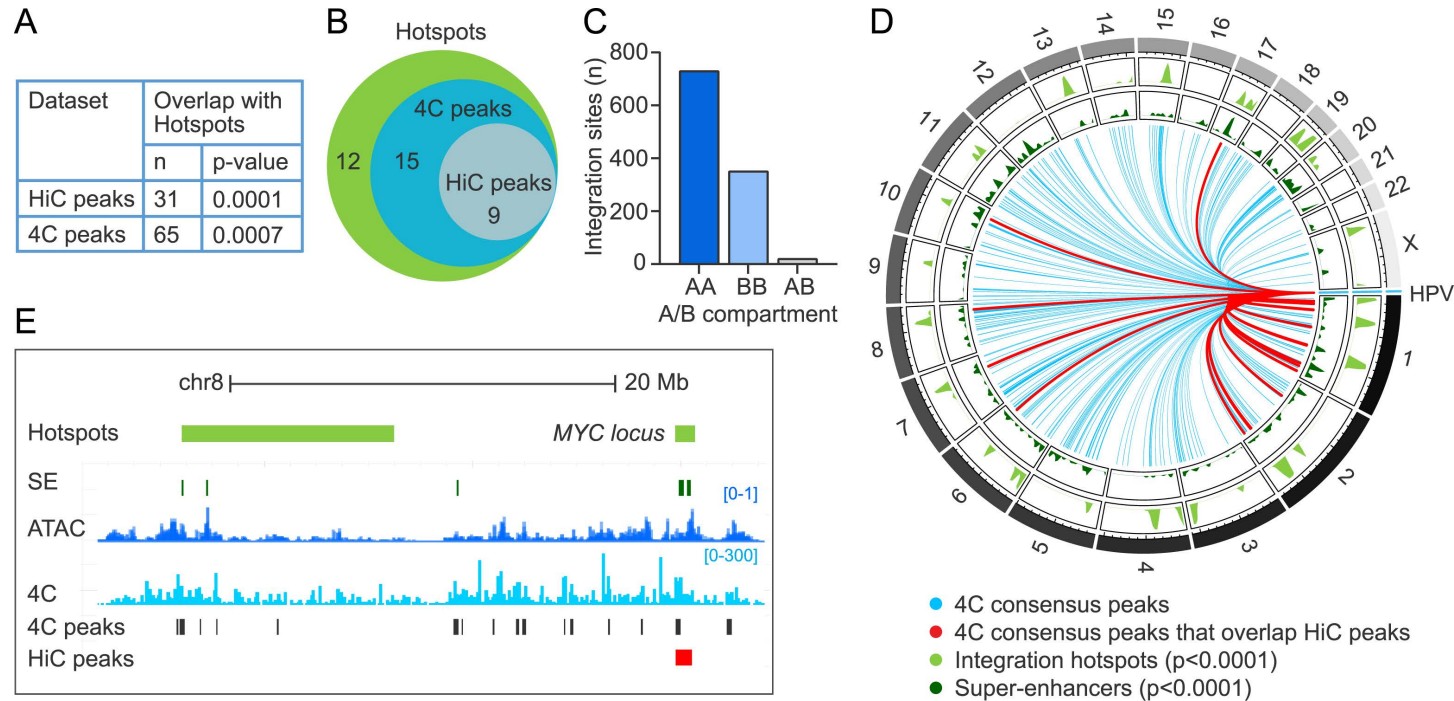

**Fig 7. Enrichment of HPV31 *trans*-interactions at HPV integration hotspots.** HPV31 association sites within the host genome were correlated with sites of recurrent integration (hotspots) defined in different cervical tumors [17]. A, Table showing the regions of overlap between HPV31 *trans*-interactions mapped through HiC (n = 87 peaks) and 4C-seq (n = 371 consensus peaks) with sites of recurrent HPV DNA integration (n = 36 hotspots). Permutation testing showed significant overlap of HiC peaks and 4C consensus peaks with integration hotspots. B, Venn diagram showing the number of integration hotspots (n = 36) that overlap with HiC peaks and 4C consensus peaks. C, Bar chart showing the association of integration breakpoints with A/B compartments grouped as indicated in the table in Fig 5D. D, Circos plot showing HPV31 association sites with host chromosomes. Blue lines represent viral-host *trans*-interactions identified by 4C-seq (consensus peaks); red lines mark consensus sites also identified by HiC (n = 28). Bootstrap analysis showed HPV31 *trans*-interactions to be significantly enriched at super-enhancers and integration hotspots. E, Alignment of viewpoint 3 4C peaks (light blue peaks) to the HPV integration hotspot (light green bars) at the MYC locus on chr8. Consensus peaks are marked by the grey bars; the red bar indicates a HiC consensus peak. Super-enhancers (SE) are indicated by dark green bars and ATAC-seq signal by dark blue peaks.

these sites to be significantly enriched at A compartments (p-value, 1.0E-4). The association of HPV31 genome interacting regions with both integration hotspots and super-enhancers is displayed in the circos plot in Fig 7D and shows a strong correlation among these genomic features. Alignment of HPV31 *trans*-interactions, ATAC-seq signal and super-enhancers at two integration hotspots at the MYC gene locus on chromosome 8 are shown in Fig 7E.

Collectively, these data show that HPV31 minichromosomes preferentially associate with transcriptionally active and open host chromatin, including regulatory domains that contain super-enhancers and regions that are frequently observed as integration sites for HPV DNA in cervical cancer.

### HPV genomes associate with genomic regions prone to double-stranded DNA breaks

Transcriptionally active regions are susceptible to double-stranded DNA breaks (DSB) that are restored by the non-homologous end-joining or homologous recombination repair pathways [50]. Moreover, papillomaviruses hijack host DNA damage responses to replicate their genomes [51]. We therefore correlated HPV31 genome association sites with DSB mapped in NHEK cells by DSBCapture, which were shown to be highly associated with transcriptionally active regions [52]. We first compared the DSBCapture dataset with our 9E ATAC-seq dataset, and H3K27ac ChIP-seq dataset that we previously mapped in W12-20863 cells [17]. Similar to nucleosome depleted regions (identified by ATAC-seq) and active

enhancers (marked by H3K27ac), and consistent with previous findings in NHEK cells [52], DSBs were highly enriched at transcriptional start sites in cervical keratinocytes, Fig 8A.

Next, we correlated the DSBCapture dataset with our HiC and 4C-seq datasets. Permutation testing showed that normalized DSBCapture read counts were significantly higher at HPV31 genome association sites relative to size matched control regions for both the HiC (p-value, 1.5E-10) and 4C-seq (p-value, < 2.2E-16) datasets, Fig 8B-C. This indicates that HPV genomes associate with regions of host chromatin that are prone to DSBs.

We show that HPV minichromosomes associate with sites susceptible to recurrent HPV DNA integration (Fig 7) and questioned whether these integration breakpoints correlated with regions of host chromatin that are susceptible to DSBs. Permutation testing showed that normalized DSBCapture read counts were significantly higher at genomic loci within 50 kb of integration breakpoints previously identified in cervical tumors (S11 Table) relative to size matched control regions (p-value, < 2.2E-16), Fig 8D. We further used permutation testing to show that previously defined BREAK-seq peaks [52] were significantly enriched at HiC peaks, 4C consensus peaks, and integration breakpoints relative to size-matched random control regions (S12 Table).

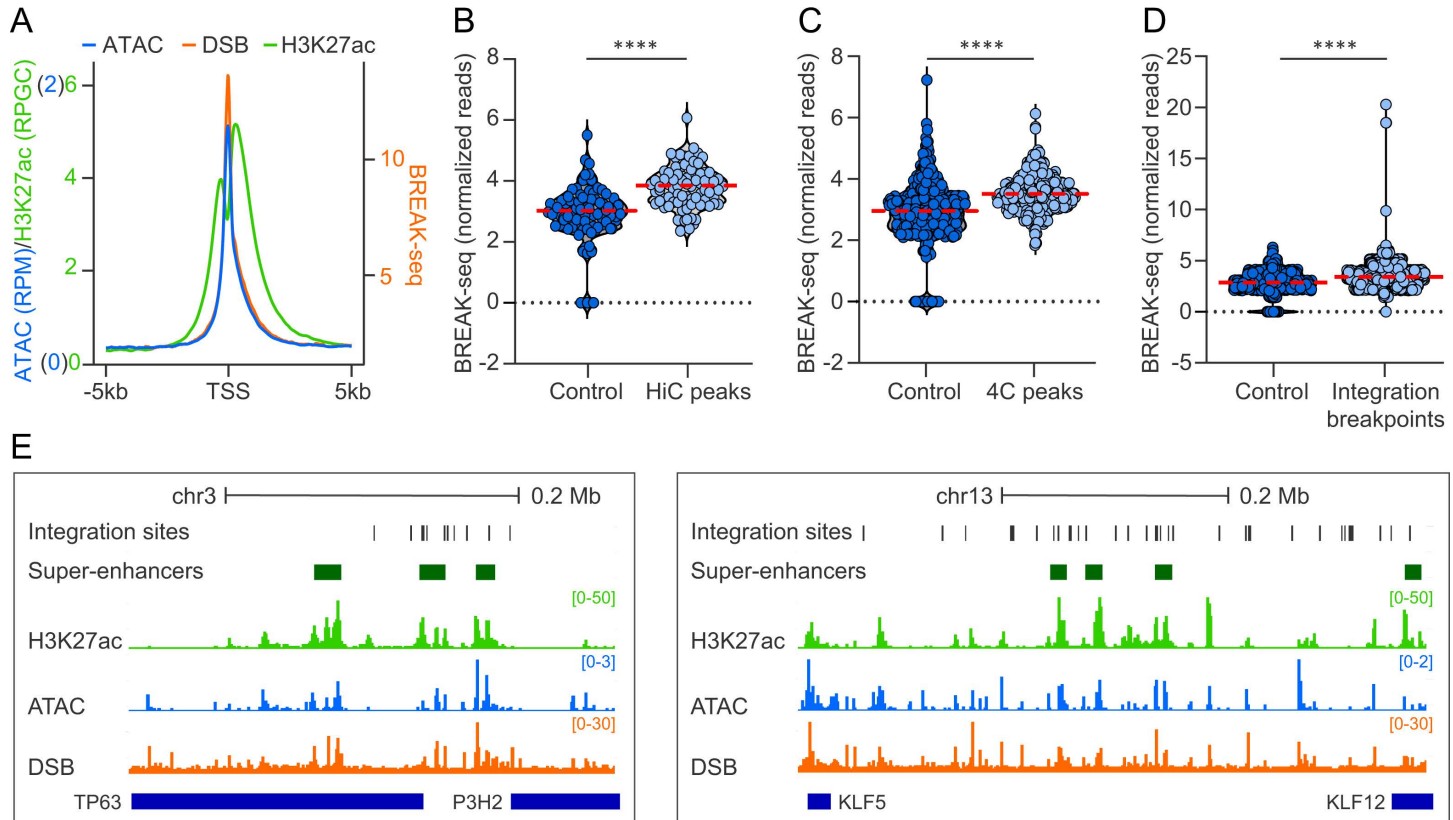

**Fig 8. HPV genomes are enriched at regions of host chromatin susceptible to double-stranded DNA breaks.** HPV31 association sites and HPV integration breakpoints were correlated with regions prone to double-stranded DNA breaks (DSB). A, Profile plot of 9E ATAC-seq, W12-20863 H3K27ac ChIP-seq [17] and NHEK DSBCapture (BREAK-seq) [52] normalized read counts at TSS (±5 kb). B-D, Violin plots showing BREAK-seq signal intensities (average normalized reads) profiled in NHEK cells at HiC peaks (B),4C consensus peaks (C) and HPV integration breakpoints ±50 kb flanks (S11 Table) (D) relative to size-matched random control regions. Red dotted line indicates mean intensity of BREAK-seq signal. Wilcoxon rank-sum test was used to determine statistical significance (****, p < 0.0001). E, Alignment of H3K27ac (green peaks), ATAC-seq (blue peaks) and BREAK-seq (DSB; orange peaks) datasets to chromosomes 3 (*left*, chr3:189,422,119-189,742,935; hg19) and 13 (*right*, chr13:73,628,066-74,292,772; hg19). HPV integration breakpoints from cervical tumors and cellular super-enhancers are denoted by dark grey and green bars, respectively.

Alignment of ATAC-seq, H3K27ac ChIP-seq, DSBCapture and super-enhancer datasets at two integration hotspots at the TP63 and KLF5/12 gene loci on chromosomes 3 and 13, respectively, are shown in Fig 8E. We conclude that association of HPV minichromosomes at regions of the host genome that are susceptible to DSBs increases the chances of accidental viral genome integration. This offers a novel explanation for the frequent integration of HPV DNA into transcriptionally active regions of host chromatin (Fig 9).

## Discussion

In this study, we used chromosome conformation capture sequencing to study genome-wide viral-host chromatin interactions during persistent HPV infection. By leveraging the unbiased nature of HiC to detect all chromosomal interactions, we effectively mapped viral genome association sites and showed preferential association of HPV31 genomes with transcriptionally active chromatin. To increase the sequencing depth required to more specifically detect viral-host *trans*-interactions, especially in cells with low viral DNA copy number, we performed 4C-seq using the HPV31 genome as bait. This corroborated the findings of HiC in that HPV31 genomes preferentially associated with active chromatin. We also used the CIN612-6E subclone, which contains a single locus of integrated HPV31 DNA, to validate the 4C assay and enable background correction for HPV31 *trans*-interactions in the 9E subclone. As expected, the 6E cells showed strong enrichment of *cis*-interactions at the HPV integration locus, consistent with previous reports for integrated HPV16 DNA [31].

We identified active and open chromatin using our previously published W12-20863 H3K27ac ChIP-seq dataset [17] and a new ATAC-seq dataset obtained from 9E cells, respectively. We also delineated A and B chromatin compartments from our HiC dataset. HPV31 genome *trans*-interactions positively correlated with each of these active chromatin regions in both the HiC and 4C datasets. This is consistent with previous studies in cervical keratinocytes showing an association between papillomavirus E2 proteins, the cellular super-enhancer marker Brd4, and transcriptionally active chromatin [7,8]. Other DNA viruses have been shown to associate with transcriptionally active regions [33,53,54], including association

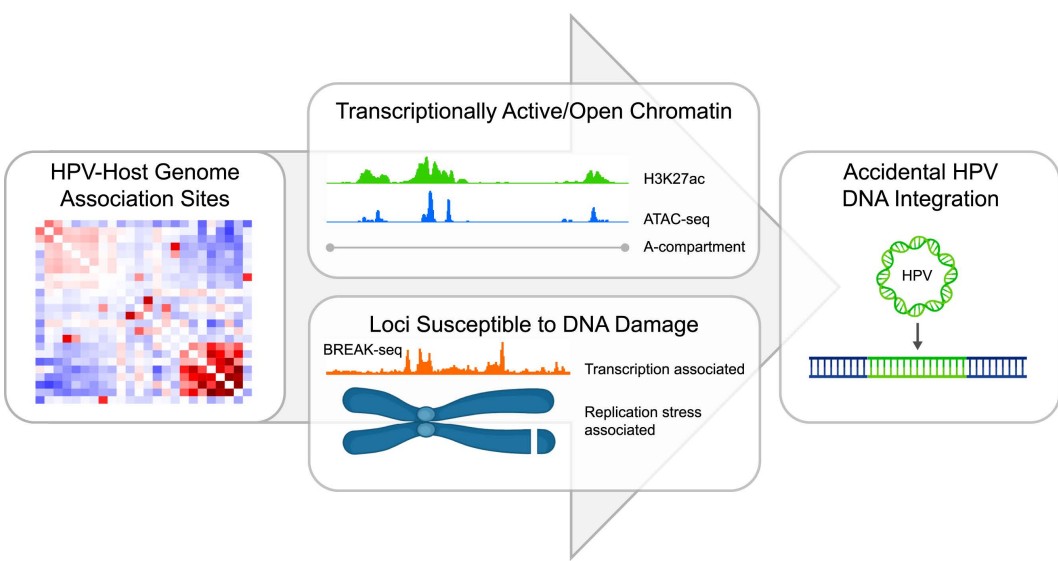

**Fig 9. Association of HPV Genomes with Regions of Host Chromatin susceptible to DNA Damage can lead to Accidental Integration.** We show that HPV minichromosomes preferentially associate with active and accessible host chromatin during maintenance replication; these regions are also prone to dsDNA breaks. We propose that this association (in addition to the previously described association with common fragile sites) increases the susceptibility of these regions to HPV genome integration. Parts of the figure were created in BioRender, https://biorender.com/86ycra4.

of EBV genomes with super-enhancers [55] in a latency type-dependent manner. Interchromosomal interactions occur at active nuclear speckles [56], where super-enhancers have also been shown to associate with [57] and concentrate the transcriptional machinery to regulate cell-identity gene expression [58].

HPV genomes are thought to tether to host mitotic chromosomes to retain and partition their genomes in dividing cells [1] and recent studies have shown random association of HPV genomes with host mitotic chromatin, often in large puncta indicative of viral genome clusters [59,60]. Like papillomaviruses, EBV associates with host mitotic chromosomes to partition their genomes in dividing cells [1] and a study investigating EBV genome attachment to host chromatin showed that viral tethering sites were maintained in mitotic and interphase cells [33]. Although we did not assess viral-host *trans*-interactions in mitotic cells, a previous study from our laboratory identified Persistent E2 and BRD4-Broad Localized Enrichments of Chromatin (PEB-BLOCS) regions, where HPV E2 and Brd4 bound constitutively throughout the cell cycle [8]. Furthermore, productive HPV31 genome amplification occurs adjacent to these regions in differentiated HPV-infected keratinocytes. During mitosis, most super-enhancers retain H3K27 acetylation to ensure rapid and faithful expression of cell-identity genes [61,62]. Thus, clustering of viral genomes to these regulatory elements offers a strategic means to ensure viral transcription and low-level persistent infection.

We show that the early region of the HPV31 genome has the highest frequency of HiC contacts with host chromatin. Recruitment of CTCF to the early region of alpha-HPVs has been shown to promote chromatin loop formation between the viral URR and E2 gene to regulate viral gene expression [63,64]. CTCF is important for modulating chromatin topology for several viruses [65] and it is intriguing to postulate that it is important for HPV genome tethering, but realistically the small size of the HPV genome may preclude precise identification of viral-host chromatin interaction by chromosome conformation capture techniques.

Previous studies have shown that HPV DNA integration frequently occurs at common fragile sites [11–13], which are regions of the genome prone to replication stress. It has been proposed that DNA damage and repair of fragile sites increase the likelihood of accidental HPV integration. Transient dsDNA breaks can occur at transcriptional regulatory elements, including super-enhancers [66,67], and a recent study showed that >90% of the most highly expressed genes in normal human epidermal keratinocytes were susceptible to dsDNA breaks at their transcriptional start sites [52]. Many cellular factors function both in viral and host transcription, replication, and DNA repair, and HPV hijacks DNA repair processes for genome replication [51]. For example, Brd4 and TopBP1 are associated with replicating HPV genomes [68–70] and are also recruited to host dsDNA breaks to promote homologous recombination and maintain genomic integrity [71,72]. Here, we show that viral-host genome interaction sites are enriched in super-enhancer elements, nucleosome-depleted regions and active chromatin in cervical keratinocytes. These transcriptionally active regions also positively correlate with DSBs mapped in NHEK. A limitation of this study is the use of HPV-negative, epidermal keratinocytes for comparison of viral genome association sites with DSBs. HPV oncoprotein expression can induce DNA breaks within the host genome [73], and 9E cells have been shown to have higher levels of DSBs relative to NHEK cells [74]. Therefore, increased DSBs in HPV infected cells could enhance the frequency of integration at these additional DNA break sites.

In this current study, we did not observe significant *trans*-interactions at the 6E integration locus site in 9E cells, which likely reflects the stochastic nature of integration and clonal selection of integrated genomes. From the literature, HPV31-positive cervical tumors represent <10% of tumors with integrated viral genomes [75–77]. However, integrated HPV31 genomes from several cervical tumors overlap integration hotspots, including those on chromosomes 3 and 17 that contain the KLF5/KLF12 [75] and KRT9/KRT14 genes [76], respectively. Furthermore, the genomes of several HPV types (HPV16, HPV18, HPV45 and HPV59) were found integrated at integration hotspots in multiple tumor samples [17,75,76], indicating that integration into these regions is common among different high-risk HPV types. We propose that HPV minichromosomes maintain an active and persistent infection by associating with host euchromatin. However, an unfortunate consequence of this association is increased susceptibility of these regions to dsDNA breaks, which could promote viral integration.

## Materials and methods

### Cell culture

CIN612-6E (RRID: CVCL_ER26) and CIN612-9E (RRID: CVCL_ER27) human-derived cervical keratinocytes were originally isolated from a cervical intraepithelial neoplasia grade 1 (CIN1) lesion by Laimins and colleagues [32,78] and contain integrated or extrachromosomal HPV31 genomes, respectively. Cells were grown in F-medium (3:1 [v/v] F-12 [Ham]-DMEM, 5% FBS, 0.4 µg/ml hydrocortisone, 5 µg/ml insulin, 8.4 ng/ml cholera toxin, 10 ng/ml EGF, 24 µg/ml adenine, 100 U/ml penicillin and 100 µg/ml streptomycin) in the presence of irradiated NIH 3T3-J2 feeder cells. NIH 3T3-J2 mouse fibroblasts were cultured in DMEM containing 10% bovine calf serum.

### HiC

A subset of previously published HiC samples from two biological replicates of CIN612-9E cervical keratinocytes, Gene Expression Omnibus series GSE98120 [18], were sequenced on the Novoseq 6000 to higher read depth (University of California San Francisco Sequencing Core) for analysis of viral-host *trans*-interactions. The *in situ* HiC protocol used for preparing these samples is published elsewhere [18,79]. Briefly, approximately five million actively dividing 9E cells were either crosslinked *in situ* (Replicate 1) or in suspension (Replicate 2) for 10 mins at room temperature with 1% formaldehyde and quenched for 5 mins at room temperature with 0.2 M glycine. Intact nuclei were permeabilized, digested with MboI and the 5′ overhangs filled in while incorporating a biotinylated nucleotide. The resulting blunt-ended fragments were ligated, the DNA sheared, and biotinylated ligation junctions captured using streptavidin Dynabeads. Sequencing libraries were then generated using the Ovation Ultralow Library System V2 (NuGEN).

### HiC data analysis

Reads were trimmed and filtered using cutadapt version 1.18. Following trimming, reads were processed using HiC-Pro version 2.11.1 [19]. Reads were aligned to a joint reference genome of hg38 autosome and sex chromosomes and HPV31 (GenBank J04353.1). The genome was digested by the HiC-Pro script digest_genome.py using an "mboi" digestion pattern. Contact maps were created using a window size of 500 kb and 1 Mb. *Trans*-interactions were called using MHiC version 0.5.0 (R/3.6.1) [20]. First, data from chrY and (mitochondrial) chrM were removed from the 1 Mb raw matrix file and matching coordinate bed file. Next, MHiC was used to call all *trans*-interactions using default parameters. MHiC identifies significant interactions by assuming a background binomial distribution model. Multiple hypothesis correction is achieved using the Benjamini-Hochberg method. Only interactions with an adjusted p-value (q-value) of less than 0.1 were retained as significant. A/B compartments were called using HiTC version 1.48.0 (R/4.4.1) [37] using the pca.hic function and a custom gene reference based upon only protein coding genes from the gencode 28 release. Heat maps of chromosome-resolution interactions were determined from the HiC-Pro contact maps and expected interactions calculated for each chromosome pair as previously described [18].

### 4C-seq

4C was performed in three and four biological replicates of 6E and 9E cells, respectively, using methods adapted from those previously described [30,80,81]. Briefly, approximately 5 million cells were crosslinked *in situ* with 2% formaldehyde for 10 min at room temperature and quenched for 5 min with 0.125 M glycine. Cells were lysed using 0.2-0.5% Igepal CA-630, centrifuged at 3,000 x g for 5 min at 4°C and the resulting nuclei permeabilized in 0.3-0.7% SDS for 1 hour at 37°C, with shaking at 900 rpm, followed by sequestration of SDS by 2% Triton X-100 for 1 hour at 37°C, 900 rpm. Nuclei were washed twice with 1X EcoRI buffer (NEB; 100 mM Tris-HCl, 50 mM NaCl, 10 mM MgCl$_2$, 0.025% Triton X-100, pH 7.5 at 25°C) to remove SDS and digested overnight with 400U EcoRI (NEB, R3101M) at 37°C, with shaking at 900 rpm. An additional 300U EcoRI was added and the nuclei incubated for a further 4 hours prior to heat inactivation at 65°C

for 20 min. Digested samples were ligated overnight in 7 ml 1X ligase buffer (NEB; 50 mM Tris-HCl, 10 mM MgCl$_2$, 1 mM ATP, 10 mM DTT, pH 7.5 at 25°C) containing 400 cohesive-end units T4 DNA ligase (NEB, M0202L) at 16°C, 500 rpm, and crosslinks reversed overnight at 56°C, 500 rpm, in the presence of 0.04 mg/ml proteinase K [80]. Samples were then incubated with 0.04 mg/ml RNase A for 30 min at 37°C, 500 rpm. For sample S14, the primary ligation was performed overnight at 16°C, 1,400 rpm, in 50 μl 1X ligase buffer (NEB), supplemented with 0.1% Triton X-100 and 670 cohesive-end units T4 DNA ligase (NEB, M0202L), and crosslinks reversed overnight at 56°C in the presence of 0.2 mg/ml proteinase K [81]. Samples were then incubated with 0.3 mg/ml RNase A for 30 min at 37°C. DNA was purified by phenol:chloro-form:isoamyl alcohol (25:24:1, v/v; Invitrogen, 15593–031) and chloroform (Macron Fine Chemicals) extractions and digested with 150U DpnII (NEB, R0543M) overnight at 37°C, with shaking at 900 rpm, prior to heat inactivation at 65°C for 20 min. Digested DNA samples were ligated and purified by phenol:chloroform and chloroform extractions, followed by an additional purification step using the QIAquick PCR Purification Kit (QIAGEN, 28106). Purified DNA was subjected to inverse PCR using the Expand Long Template PCR System (Roche, 11681842001) and primers listed in S2 Table, which target three viewpoints within the HPV31 genome (Fig 2B). PCR products for the three viewpoints were pooled for each sample, purified using the QIAquick PCR Purification Kit (QIAGEN, 28106) and subjected to 2 x 100bp paired-end read sequencing on the NovaSeq 6000 platform (Illumina Genome Network) to a sequencing depth of >320 million read pairs per sample (S3 Table). Sequencing libraries were generated using the Truseq DNA Nano kit (Illumina) and subjected to size selection to remove fragments >1,000 bp that typically represent undigested or self-ligated products. To ensure high sequence diversity of the 4C-seq libraries, a PhiX spike-in control of 30% the sequencing run was included. Sequencing was performed by the Frederick National Laboratory for Cancer Research (FNLCR) Sequencing Facility (National Cancer Institute/National Institutes of Health).

## 4C-seq data analysis

The pipe4C pipeline was used to map viral-host genome contact profiles for the three viewpoint reading primers listed in S2 Table. R1 and R2 reads were processed independently using pipe4C version 1.1.6 [30]. Reads were aligned to a joint reference genome of hg38 autosome and sex chromosomes and HPV31. Read counts per fragment in the pipe4C output files were combined for R1 and R2. Normalized read counts were calculated using the same steps as in the pipe4C code. Specifically, read counts in genomic regions were normalized to reads per million, excluding the top highest two values to mitigate outliers. Normalized counts were then collapsed into 50 kb windows using GenomicRanges and BSgenome. Hsapiens.UCSC.hg38. Pearson correlation coefficient plots were generated using ComplexHeatmap [82].

## 4C-seq *trans*-interaction peak calling

Fifty kb-binned bedgraph files were imported for all viewpoint 3 data using rtracklayer and are available at GEO accession GSE294036. Chromosome 4 was filtered out from the 6E samples using GenomicRanges to remove bias of the *cis*-interactions on that chromosome. The mean value of the normalized reads across all samples was calculated and used as the lambda parameter for a Poisson distribution. Based upon this Poisson distribution, p-values were calculated for the binned data for each sample using the ppois function and transformed to -log10 values. For each sample, regions with a score greater than five (indicating statistical significance) were extracted, merged using a maximum gap size of 50 kb, and filtered to retain regions longer than 50 kb. Consensus peaks were defined as significant HPV31 *trans*-interactions found in at least three out of four 4C biological replicates.

## ATAC-seq

CIN612-9E cervical keratinocytes were seeded onto 10 cm plates of irradiated mouse J2/3T3 fibroblasts at 6x10e5 cells per plate and cryopreserved in F-media supplemented with 5% DMSO two days later, after removal of the fibroblasts. The assay for transposase-accessible chromatin by sequencing (ATAC-seq) was performed by the CCR Sequencing Facility

(National Cancer Institute/National Institutes of Health) using the Omni-ATAC protocol [83]. ATAC samples (two biological replicates) were pooled and subjected to 2 x 100 bp paired-end read sequencing on the Illumina NextSeq 2000 P2 platform. All the samples had Q30 bases above 93% and yields between 103 and 141 million pass filter reads. Reads were trimmed for adapters using Cutadapt v1.18 and aligned to the human hg19 reference genome using Bowtie 2 v2.3.4.1 with flag -k 10 [84]. Peaks were called using Genrich v0.6 (https://github.com/jsh58/Genrich) with the following flags: -j -y -r -v -d 150 -m 5 -e chrM, chrY. Genrich-produced bedgraphs were normalized by library size (reads per million sequenced reads, RPM) for visualization. Genomic coordinates of HiC and 4C peaks were converted from human reference genome hg38 to hg19 using LiftOver, https://genome.ucsc.edu/cgi-bin/hgLiftOver [85] for alignment with ATAC-seq datasets.

## Overlap analysis of HPV31 *trans*-interactions

Intersect between genomic coordinates of *trans*-interactions identified by HiC (S1 Table) and 4C-seq (S6 Table) with each other or different genomic features detailed below was performed using bedtools Intersect intervals and default parameters [86]. Super-enhancer (S8 Table) and integration hotspot (S10 Table) datasets were taken from our previous study [17], and the genomic coordinates converted from human reference genome hg19 to hg38 using LiftOver, https://genome.ucsc.edu/cgi-bin/hgLiftOver [85]. For significance testing, data were permutated 10,000 times to create an expected distribution of overlap between *trans*-interactions and different genomic features relative to random size-matched control regions using regioneR [87]. Note that the maximum statistical significance attainable with 10,000 permutations corresponds to a p-value of 1.0E-4.

## Overlap analysis with double-stranded DNA breaks

NHEK DSBCapture BREAK-seq (GSM2068755 and GSM2068756) and 20863 H3K27ac ChIP-seq (GSM5550313 and GSM5550314) datasets are available from GEO accession series GSE78172 and GSE183048, respectively. For profile plots, EnrichedHeatmap version 1.34 was used to extract the TSS sites from the ATAC-seq, BREAK-seq and H3K27ac ChIP-seq datasets and the data smoothed across 25 bp windows before plotting. For significance testing, the mean normalized reads across all regions of interest within a given dataset (HiC peaks, S1 Table; 4C consensus peaks, S6 Table or HPV integration breakpoints, S11 Table) were compared against the mean normalized reads of size-matched random control regions using the Wilcoxon rank-sum test. ENCODE hg19 v2 blacklist regions [88] were excluded from this analysis.

## Chromosomal ideograms

Ideograms displaying viral-host *trans*-interactions across each chromosome were generated using PhenoGram (https://visualization.ritchielab.org/phenograms/plot) and karyoploteR [89].

## Gene ontology analysis

4C consensus peaks (n = 371) were analyzed against the human hg38 reference genome using the Genomic Regions Enrichment of Annotations Tool (GREAT), version 4.0.4, accessed February 2025 from https://great.stanford.edu/great/public-4.0.4/html/index.php. The default Basal Plus Extension gene regulatory domain definition was used for association analysis of 4C consensus peaks with host genes. This definition assigns a minimum distance of 5 Kb upstream and 1 Kb downstream of a TSS (regardless of other neighboring genes) as the basal regulatory domain of each gene. The gene regulatory domain is extended in both directions to the basal domain of the nearest gene up to a maximum extension of 1 Mb in one direction.

## Circos plots

Significant viral-host chromosomal interactions identified by 4C-seq were plotted using the R package circlize [90].

## Polymerase chain reaction (PCR)

Total genomic DNA was purified from 2 x 10⁶ CIN612 cells using the DNeasy Blood and Tissue kit (Qiagen) and subjected to PCR analysis using primers listed in S13 Table. The FastStart Taq DNA Polymerase kit (Roche) was used to amplify DNA fragments <3 Kb, and the Expand Long Template PCR System (Roche) for amplicons >3 Kb. PCR products were separated on 0.8% TAE agarose gels containing 0.5 µg/ml ethidium bromide (Invitrogen).

## Supporting information

**S1 Fig.  Regions of the HPV31 genome that contact host chromatin.** Viral-host trans-interactions mapped by HiC were aligned to the linearized HPV31 reference genome (GenBank J04353.1). The 5' and 3' read pairs associated with each of the indicated MboI restriction sites were plotted as the percentage of reads at each cut site relative to the total number of reads across the HPV31 genome for both HiC datasets (Replicates 1 and 2). Viral-host contacts were filtered to only include reads where the mate pair on the human genome localized to a significant 1 Mb bin listed in S1 Table. Sequencing reads that did not align to a MboI restriction site were excluded from this analysis.
(PDF)

**S2 Fig.  Deletions within the integrated hpv31 genome in 6e cells detected by pcr analysis.** In all three 6E samples, the primary restriction site associated with the 5' end of the trimmed sequencing reads for viewpoint 2 were not detected, and so no output files were generated. For viewpoint 1, the primary restriction site was only detected in one of the read pairs generated by paired-end sequencing in all 6E samples. This suggested that integration of the HPV31 genome had disrupted the region targeted by the 4C viewpoint 1 and 2 primers in 6E cells. To test this, gENOMIC DNA ISOLATED FROM 6E AND 9E CELLS WAS SUBJECTED TO PCR ANALYSIS USING primer pairs that span approximately every 500 bp of the HPV31 genome. A, HPV31 GENOME SHOWING THE REGIONS TARGETED BY PCR PRIMER PAIRS (GREY BARS LABELLED P1-P17 THAT CORRESPOND TO THE AMPLICONS SHOWN IN PANEL B; SEE S13 Table FOR PRIMER SEQUENCES); RED FONT INDICATES REGIONS DELETED IN 6E CELLS BASED ON PCR ANALYSIS (B). B, PCR products for primer pairs 1–17, indicted in panel A, were analyzed by gel electrophoresis. PCR analysis of genomic DNA isolated from 6E and 9E cells identified large deletions within the viral E1, E2, L2 and L1 genes in 6E cells, including the regions targeted by the 4C viewpoint 1 and 2 non-reading primers. V1-V3, 4C Viewpoints; NTC, no template control.
(PDF)

**S1 Table.  Significant HPV31 trans-interactions mapped in CIN612-9E cells by HiC.**
(XLSX)

**S2 Table.  4C Inverse PCR primers targeting the HPV31 genome.**
(XLSX)

**S3 Table.  4C sample details.**
(XLSX)

**S4 Table.  HPV31 trans-interactions mapped in CIN612-6E cells by 4C.**
(XLSX)

**S5 Table.  Significant HPV31 trans-interactions mapped in CIN612-9E cells by 4C.**
(XLSX)

**S6 Table.  HPV31 4C consensus peaks defined in CIN612-9E cells.**
(XLSX)

**S7 Table. A/B compartments defined in two HiC datasets from CIN612-9E cells using HiTC [Servant, 2012].**
(XLSX)

**S8 Table. Super-enhancers profiled in W12 HPV16-positive cervical keratinocytes [Warburton, 2021].**
(XLSX)

**S9 Table. Enriched Biological Processes associated with 4C consensus peaks identified using GREAT analysis [McLean, 2010].**
(XLSX)

**S10 Table. Integration hotspots defined in CESC [Warburton, 2021].**
(XLSX)

**S11 Table. Integration breakpoints (-/+ 50 kb flanking regions) defined in CESC [Warburton, 2021].**
(XLSX)

**S12 Table. BREAK-seq peaks mapped in NHEK [Lensing, 2016].**
(XLSX)

**S13 Table. PCR primers spanning the HPV31 genome.**
(XLSX)

## Acknowledgments

We thank Stephanie A. Moquin (University of California, San Francisco and Gladstone Institute of Virology and Immunology) for preparing the original HiC libraries, Laimonis Laimins (Northwestern University) for the CIN612-6E and -9E cells, Justin Lack (Research Technologies Branch/NIAID/NIH) for advice on sequencing and statistical analyses, and Asya Khleborodova (Research Technologies Branch/NIAID/NIH) for processing ATAC-seq datasets. This work utilized the computational resources of the NIH HPC Biowulf cluster, http://hpc.nih.gov.

## Author contributions

**Conceptualization:** Alix Warburton, JJ L. Miranda, Alison Anne McBride.

**Data curation:** Alix Warburton, Tovah E Markowitz, JJ L. Miranda.

**Formal analysis:** Alix Warburton, Tovah E Markowitz, JJ L. Miranda, Alison Anne McBride.

**Funding acquisition:** Alison Anne McBride.

**Investigation:** Alix Warburton, Tovah E Markowitz, Alison Anne McBride.

**Methodology:** Alix Warburton, Tovah E Markowitz, JJ L. Miranda, Kinjal Majumder.

**Project administration:** JJ L. Miranda, Alison Anne McBride.

**Resources:** Alison Anne McBride.

**Software:** Tovah E Markowitz.

**Supervision:** Alison Anne McBride.

**Validation:** Alix Warburton, Tovah E Markowitz.

**Visualization:** Alix Warburton, Tovah E Markowitz.

**Writing – original draft:** Alix Warburton, Alison Anne McBride.

**Writing – review & editing:** Alix Warburton, Tovah E Markowitz, JJ L. Miranda, Kinjal Majumder, Alison Anne McBride.

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
