## [Decision Letter · Decision Letter 0]

17 Jun 2025

Human Papillomavirus Genomes Associate with Active Host Chromatin during Persistent Viral Infection

PLOS Pathogens

Dear Dr. McBride,

Thank you for submitting your manuscript to PLOS Pathogens. After careful consideration, we feel that it has merit but does not fully meet PLOS Pathogens's publication criteria as it currently stands. Therefore, we invite you to submit a revised version of the manuscript that addresses the points raised during the review process.

Please submit your revised manuscript within 60 days Aug 16 2025 11:59PM. If you will need more time than this to complete your revisions, please reply to this message or contact the journal office at plospathogens@plos.org. Please include the following items when submitting your revised manuscript:

We look forward to receiving your revised manuscript.

Kind regards,

Cary A. Moody

Academic Editor

PLOS Pathogens

Blossom Damania

Section Editor

PLOS Pathogens

Editor-in-Chief

PLOS Pathogens

orcid.org/0000-0003-2946-9497

Editor-in-Chief

PLOS Pathogens

orcid.org/0000-0002-7699-2064

**Journal Requirements:**

**Reviewers' Comments:**

Reviewer's Responses to Questions

**Part I - Summary**

Reviewer #1: Warburton et al perform HiC and 4C analyses on cells with HPV 31 episomes and show that viral genomes often associate with transcriptionally active compartments in open chromatin regions. Using ChiP-seq for Brd4 and H3K27ac the authors report a linkage to regions containing super enhancers. Additional analyses examine published double strand BREAK data sets from normal cells to conclude that the sites of association are often linked to sites of viral genome association. This suggests a potential linkage of viral integration sites to regions susceptible to double strand DNA break formation. This is an interesting and detailed bioinformatic analysis of sites at which HPV genomes associate with host chromatin and potential integration sites. The manuscript is densely written and could benefit from expanded discussion of the points discussed below. In the second part of their study, the authors make comparisons between different cell lines including normal NHEKs and HPV positive which this needs to be described more fully and the implications of using such lines discussed in more detail. Several points should be addressed:

Reviewer #2: The manuscript by Warburton and colleagues use Hi-C and 4-C sequencing of the HPV31 Cervical keratinocyte line CIN612-9E to identify host chromosome association with a variety of other genomic methods. They conclude that viral genomes associate with active transcription regions. While they claim that this serves to maintain active viral gene expression, this was not tested. A limitation of the study is the use of genomic datasets from other cell lines instead of the 9E line.

Reviewer #3: In this manuscript, Warburton et al. use chromosome conformation capture assays (Hi-C and 4C-Seq) to map the association of the HPV31 genome—either in its extrachromosomal or integrated form—with host chromatin. Their findings regarding the integrated form of the viral genome are consistent with previous studies based on curated dataset analyses or cell lines containing integrated HPV genomes (Warburton, Genomic Medicine, 2021; Singh, Molecular Oncology, 2023), showing that the viral genome is associated with active chromatin regions without a specific chromosome preference.

An important finding of this study is the association of the extrachromosomal genome with transcriptionally active regions (as shown by ATAC-Seq experiments and previously published datasets for Brd4 and H3K27Ac ChIP-seq), HPV integration hotspots (from previously published HPV integration datasets), and regions prone to double-strand breaks (from DSBCapture datasets). While the authors report that the association of the extrachromosomal viral genome is not limited to a specific region of the viral genome, the intersection of Hi-C and 4C-Seq data identified 19 regions of the human genome—including MYC, E2F2, and CASP8—that are common across analyses. These data support a model in which the extrachromosomal HPV genome is enriched at specific genomic locations.

These findings are significant as they reveal a novel correlation between the localization of the extrachromosomal form of HPV and integration sites. Minor changes are discussed below.

**Part II – Major Issues: Key Experiments Required for Acceptance**

Reviewer #1: 1. What is the difference between SM159 and SM163? In Figure 1C there are significant differences in viral-host trans interactions between the two cell lines. What accounts for these differences? Could these interactions be dynamic and change as a function of passage? This needs to be stated more directly in the text as it is only evident when examining the figure. The implication of this variability with different isolates on their conclusions needs to be discussed in detail.

2. (Figure 8) The ATAC-seq, H3K27ac/Brd4 ChIP-seq, and DSBcapture are all performed on different cell lines (9e, W12, and NHEKs, respectively). Is this the best comparison? These assays are admittedly expensive and time consuming and may be the reason for the choices but particularly the use of NHEKs alone to look at DNA breaks is not ideal. W12 and CIN 612 are more closely aligned.

3. The authors use a published DSBcapture dataset from NHEKs which are normal cells. The levels of DNA breaks are increased in CIN 612 cells relative to normal cells and the use of DSBcaptuire with this HPV positive line could strengthen the conclusions. Viral-host trans interactions maybe enhanced by the induction of higher levels of DNA breaks induced by viral proteins.

4. (Figure 8B-D) Increased DSBcapture reads were observed in the 50kb flanking regions from the called HiC, 4C, and integration breakpoint peaks. Do these increased DSBcapture reads correlate with the called DSBcapture peaks from Lensing el al, 2016? The y-axes of the figure 8B-D is not clearly labeled and this needs to be adjusted (Read counts vs read counts normalized to input, etc).

Reviewer #2: The major issue in this manuscript is the co-mingling of so many diverse datasets from a variety of cell lines. While this might not be an issue if every line tested behaves similarly whether it has HPV16 or HPV31, if the viral genome is integrated or episomal. Are the super enhancers identified by ChIP seq of HPV16 keratinocytes going to be the same in the 9E cells? It certainly is possible but this issue should be addressed . Similarly, are the DSBs identified in NHEK cells relevant to the 9E cells. Again, they might be relevant but the question was not addressed.

Reviewer #3: No major issue has been found.

**Part III – Minor Issues: Editorial and Data Presentation Modifications**

Reviewer #1: (No Response)

Reviewer #2: Fig2C. It seems odd that all the correlations among the viewpoints are between 0.61 and 0.64. Was there any biases in the calculations?

Line 180-181. Why do you assume that all of the 4C-seq interactions with other chromosomes are background? Is it possible that the integrated viral genome could connect with other transcriptional active regions on other chromosomes?

Line 206 why not illustrate the connections between HPV and MYC, E2F2, and CASP8 similar to Figure 3?

Line 244-246. Fig 5G. How was this region of the genome selected for illustration? Are there any relevant genes in this region? Myc? E2F2? CASP8?

Line252-255. Figure 6, how sure are you that the super enhancers from a different cell line are relevant?

Figure 7. Is there any specific information about integration of HPV31 in cervical cancer or does every high risk genotype have similar hotspots?

Reviewer #3: 1. Figure 1 presents data on the association of extrachromosomal HPV with human chromosomes using Hi-C in CIN912 9E. Since the authors later show that HPV31 is integrated into chromosome 4 in 6E, could the authors comment on the absence of association with this analysis (1D)? Additionally, could the authors explain the lower number of associations of the HPV genome with the human genome observed in SM159 compared to SM163 in this figure?

2. The results obtained using Hi-C to map the regions of association on the HPV genome are somewhat confusing. As the authors mention the limited reliability of this technique for such analysis, these data might be better suited for inclusion in a supplementary figure, allowing the main focus to remain on the 4C-Seq analysis.

3. Since the interaction between Brd4 and E2 is required for tethering the HPV genome to daughter cells, the observed interaction of the extrachromosomal viral genome with Brd4-enriched regions is expected. Could the authors comment on any potential bias introduced by this tethering in their analyses? For instance, were most cells in the same phase of the cell cycle during the Hi-C and 4C-Seq experiments?

PLOS authors have the option to publish the peer review history of their article (what does this mean? ). If published, this will include your full peer review and any attached files.

**Do you want your identity to be public for this peer review?** For information about this choice, including consent withdrawal, please see our Privacy Policy .

Reviewer #1: No

Reviewer #2: No

Reviewer #3: No

**Figure resubmission:**

**Reproducibility:**



---

## [Decision Letter · Decision Letter 1]

30 Jul 2025

PPATHOGENS-D-25-01004R1

Human Papillomavirus Genomes Associate with Active Host Chromatin during Persistent Viral Infection

PLOS Pathogens

Dear Dr. McBride,

Thank you for submitting your manuscript to PLOS Pathogens. After careful consideration, we feel that it has merit but does not fully meet PLOS Pathogens's publication criteria as it currently stands. Therefore, we invite you to submit a revised version of the manuscript that addresses the points raised during the review process.

Please submit your revised manuscript within 30 days Sep 28 2025 11:59PM. If you will need more time than this to complete your revisions, please reply to this message or contact the journal office at plospathogens@plos.org. Please include the following items when submitting your revised manuscript:

We look forward to receiving your revised manuscript.

Kind regards,

Cary A. Moody

Academic Editor

PLOS Pathogens

Blossom Damania

Section Editor

PLOS Pathogens

Sumita Bhaduri-McIntosh

Editor-in-Chief

PLOS Pathogens

orcid.org/0000-0003-2946-9497

Michael Malim

Editor-in-Chief

PLOS Pathogens

orcid.org/0000-0002-7699-2064

**Reviewers' Comments:**

Reviewer's Responses to Questions

**Part I - Summary**

Reviewer #1: In the revised version of this manuscript, the authors have addressed and clarified many of the points raised in the initial review. The manuscript provides important insights regarding interactions between HPV genomes and host chromosomal sites through association with open chromatin, super enhancers and DNA breaks.

Reviewer #2: The revised manuscript from Warburton and colleagues demonstrate that HPV31 genomes preferentially associate with transcriptionally active A compartments of host chromatin, regions of open chromatin defined by ATAC-seq, super-enhancers defined by Brd4 and H3K27ac ChIP-seq and double-stranded DNA breaks associated with transcriptionally active sites. These association sites viral genome association sites were highly correlated with genomic loci previously identified as common HPV integration sites in cervical cancers.

Reviewer #3: Although the analyses in this study remain based on data from different cell lines, the authors now acknowledge this limitation and provide clearer context for readers to better assess the findings. The authors addressed most of the minor concerns raised by the reviewers, improving the manuscript’s readability.

**Part II – Major Issues: Key Experiments Required for Acceptance**

Reviewer #1: No major issues

Reviewer #2: No new experiments required

Reviewer #3: (No Response)

**Part III – Minor Issues: Editorial and Data Presentation Modifications**

Reviewer #1: In response to questions about differences in HPV 31 trans-interactions with various host chromosome bins (Figure 1) between Replicate 1 and Replicate 2, the authors suggest this could be due to differences in fixation methods. Equally plausible is the possibility that these interactions are dynamic and, aside from associations with super enhancers, these interactions could change with cell cycle, passage or cell clone. It might be useful to mention this possibility.

Reviewer #2: Revised manuscript has Mae several changes to clarify the use of divergent genomic databases

Reviewer #3: (No Response)

PLOS authors have the option to publish the peer review history of their article (what does this mean? ). If published, this will include your full peer review and any attached files.

**Do you want your identity to be public for this peer review?** For information about this choice, including consent withdrawal, please see our Privacy Policy .

Reviewer #1: No

Reviewer #2: No

Reviewer #3: No

**Figure resubmission:**
---

## [Editor Report · Decision Letter 2]

13 Aug 2025

Dear Dr. McBride,

We are pleased to inform you that your manuscript 'Human Papillomavirus Genomes Associate with Active Host Chromatin during Persistent Viral Infection' has been provisionally accepted for publication in PLOS Pathogens.

Best regards,

Cary A. Moody

Academic Editor

PLOS Pathogens

Blossom Damania

Section Editor

PLOS Pathogens

Sumita Bhaduri-McIntosh

Editor-in-Chief

PLOS Pathogens

orcid.org/0000-0003-2946-9497

Michael Malim

Editor-in-Chief

PLOS Pathogens

orcid.org/0000-0002-7699-2064
---

## [Editor Report · Acceptance letter]

Dear Dr. McBride,

We are delighted to inform you that your manuscript, "Human Papillomavirus Genomes Associate with Active Host Chromatin during Persistent Viral Infection," has been formally accepted for publication in PLOS Pathogens.

Best regards,

Sumita Bhaduri-McIntosh

Editor-in-Chief

PLOS Pathogens

orcid.org/0000-0003-2946-9497

Michael Malim

Editor-in-Chief

PLOS Pathogens

orcid.org/0000-0002-7699-2064